# L$^2$M$^3$OF: A Large Language Multimodal Model for Metal-Organic Frameworks

## Abstract

Large language models (LLMs) have demonstrated remarkable reasoning capabilities across diverse natural language tasks. However, comparable breakthroughs in scientific discovery are more limited, because understanding complex physical phenomena demands multifaceted representations far beyond language alone. A compelling example is the design of functional materials such as metal-organic frameworks (MOFs) – critical for a range of impactful applications like carbon capture and hydrogen storage. Navigating their vast and intricate design space in language-based representations interpretable by LLMs is challenging due to the numerous possible three-dimensional atomic arrangements and strict reticular rules of coordination geometry and topology. Despite promising early results in LLM-assisted discovery for simpler materials systems, MOF design remains heavily reliant on tacit human expertise rarely codified in textual information alone. To overcome this barrier, we introduce L$^2$M$^3$OF, the first multimodal LLM for MOFs. L$^2$M$^3$OF integrates crystal representation learning with language understanding to process structural, textual, and knowledge modalities jointly. L$^2$M$^3$OF employs a pre-trained crystal encoder with a lightweight projection layer to compress structural information into a token space, enabling efficient alignment with language instructions. To facilitate training and evaluation, we curate a structure–property–knowledge database of crystalline materials and benchmark L$^2$M$^3$OF against state-of-the-art (SOTA) closed-source LLMs such as GPT-5, Gemini-2.5-Pro, and DeepSeek-R1. Experiments show that L$^2$M$^3$OF outperforms leading text-based closed-source LLMs in property prediction and knowledge generation tasks, despite using far fewer parameters. These results highlight the importance of multimodal approaches for porous crystalline material understanding and establish L$^2$M$^3$OF as a foundation for next-generation AI systems in materials discovery.

## 1 Introduction

Metal-organic frameworks represent a versatile class of porous crystalline materials with high tunability and broad physical properties that promise transformative applications in direct carbon capture (Rohde et al., 2024), clean hydrogen storage (Chen et al., 2020), water harvesting (Alawadhi et al., 2024), and controlled drug delivery (Wu & Yang, 2017). MOF functional design involves intricate reticular synthesis procedures by linking metal atoms and organic molecules into repeating patterns, akin to 'LEGO building' at the nanoscale. Scaling-up their design is nevertheless non-trivial, even with machine learning, both due to the large number of possible building-block combinations that give rise to an enormous design space, and because of the expertise-driven nature of the design which heavily relies on domain knowledge  (Yaghi et al., 2003).

Large language models have recently emerged as powerful AI assistants for chemists, demonstrating strong reasoning capabilities in language-related chemistry tasks (Guo et al., 2023; Liu et al., 2023), such as chemical knowledge integration and tool orchestration, offering promising potential in accelerating the exploration of large design spaces (Mirza et al., 2025). The discovery of new functional materials such as MOFs however, is fundamentally more challenging because unimodal textual representations typically fail to capture complex, high-dimensional reticular phenomena that give rise to different functionalities. Unlike molecules (Zhu et al., 2024) or proteins (Wu et al., 2023),

Table 1: Model features of LLMs for crystalline materials. 'Structure' corresponds to structure prediction or structure extraction; 'Property' means property prediction; 'Knowledge' means knowledge generation, and 'Q&A' stands for question and answering.

| Model | CrystalType | LLM | Model Input | | Downstream Tasks | | | |
|---|---|---|---|---|---|---|---|---|
| | | | Text | Multimodal | Structure | Property | Knowledge | Q&A |
| LLM-Prop (Niyongabo Rubungo et al., 2025) | Inorganic | T5 | ✓ | | | ✓ | | |
| CrystLLM (Antunes et al., 2024) | Inorganic | Llama-2 | ✓ | | ✓ | | | |
| CrysText (Mohanty et al., 2024) | Inorganic | Llama-3.1 | ✓ | | ✓ | | | |
| Mat2Seq (Yan et al., 2025) | Inorganic | GPT | ✓ | | ✓ | | | |
| MatText (Alampara et al., 2025a) | Inorganic | Llama-2 | ✓ | | | ✓ | | |
| CrystalICL (Wang et al., 2025b) | Inorganic | Llama-2 | ✓ | | ✓ | | | |
| CSLLM (Song et al., 2025) | Inorganic | Llama-3 | ✓ | | | | ✓ | |
| deCIFer (Johansen et al., 2025) | Inorganic | Transformer | ✓ | | ✓ | | | |
| MatterGPT (Wang et al., 2025a) | Inorganic | GPT | ✓ | | ✓ | | | |
| Text2Struc (Baibakova, 2025) | Inorganic | CodeGen | ✓ | | ✓ | | | |
| Matterchat (Tang et al., 2025) | Inorganic | Mistral | ✓ | ✓ | | ✓ | | ✓ |
| Chemeleon (Park et al., 2025) | Inorganic | BERT | ✓ | ✓ | ✓ | | | |
| MOFGPT (Badrinarayanan et al., 2025) | MOFs | GPT | ✓ | | ✓ | ✓ | | |
| ChatMOF (Kang & Kim, 2024) | MOFs | GPT | ✓ | | ✓ | ✓ | | |
| L$^2$M$^2$OF | MOFs | Qwen2.5 | ✓ | | ✓ | ✓ | ✓ | ✓ |
| L$^2$M$^3$OF | MOFs | Qwen2.5 | ✓ | ✓ | ✓ | ✓ | ✓ | ✓ |

which can be expressed as textual sequences of a relatively small range of elements, MOFs inhabit three-dimensional, periodic structures that resist straightforward representation. In addition to being compositionally much broader, the structure-function problem for MOFs is inherently more complex because the 3-dimensional atomic 'sequence' does not encode function alone; rather, it emerges from a combination of factors, including local bonding environments, long-range crystallographic symmetry, pore connectivity, and other topological features (Luo et al., 2024).

Despite the emerging line of research on LLMs for accelerated materials design (Kang et al., 2025; Duan et al., 2025) spanning a broad range of downstream tasks, including property prediction (Niyongabo Rubungo et al., 2025) and de-novo structure generation (Wang et al., 2025a), existing approaches remain restricted to text-centric or file-based representations, such as crystallographic information files (CIFs) and text-based property descriptions (Tang et al., 2025). While effective for sequential reasoning, such encodings fail to capture three-dimensional symmetries, periodicity, and long-range structural correlations that underpin crystalline behavior, often underperforming when compared with geometry- or symmetry-aware models (Alampara et al., 2025b). A collection of existing LLMs and their modeling capacity for crystalline materials is presented in Table 1.

The challenge here extends beyond structural representation; it lies in the 'machine understanding' of materials' functionality. Multimodal integration in learning strategies, that is, leveraging atomic information as well as literature knowledge to interlink structure with function, is therefore key to enable a holistic understanding of materials' applicability. Whereas molecular modeling has seen initial success in coupling LLMs with graph neural networks or generative models (Jablonka et al., 2024), analogous strategies for crystalline systems remain rare, owing to system and design complexity, as well as the lack of standardized datasets and benchmarks tailored to crystalline materials, rendering rigorous evaluation and reproducibility quite challenging.

This work proposes L$^2$M$^3$OF, the first *multimodal LLM for MOF design* that combines multimodal MOFs representations (Park et al., 2023) with curated domain-knowledge from MOFs literature. L$^2$M$^3$OF is versatile and inherently designed to be lightweight to allow for an efficient alignment with language instructions, demonstrating SOTA performance on diverse design-critical tasks including property prediction and material application recommendation, rendering it an indispensable AI-assistant for chemists and materials scientists. To train and test L$^2$M$^3$OF, we curate the first-ever *structure-property-knowledge MOFs database*, namely MOF-SPK, featuring structural, property and domain-knowledge information for more than 100,000 MOFs materials. L$^2$M$^3$OF outperforms leading commercially-available LLMs such as DeepSeek, GPT-4o, and Gemini-2.5-Pro, demonstrating SOTA capabilities not only in capturing essential representational aspects of complex MOFs systems, but also a holistic understanding of their broader functional role and potential applicability. Fig. 1 illustrates the core architectural features of L$^2$M$^3$OF.

## 2 BACKGROUND AND RELATED WORK

**Crystal representation.** A crystal structure is defined by the geometric arrangement of its atoms within a unit cell. The unit cell represents the smallest repeating block that captures and maintains the complete symmetry and structure of the crystal. A crystal can therefore be uniquely represented as $\mathcal{M} = (\mathcal{A}, \boldsymbol{X}, \boldsymbol{L})$ using the following three parameters that characterize its unit cell: i) atom identities $\mathcal{A} = \{a_0, ..., a_N\} \in \mathbb{A}^N$, where $\mathbb{A}$ denotes the set of all chemical elements, ii) Cartesian coordinates of atoms $\boldsymbol{X} = [\boldsymbol{x}_0, ..., \boldsymbol{x}_N]^T \in \mathbb{R}^{N \times 3}$ and iii) the lattice matrix that describes the periodicity of the crystal $\boldsymbol{L} = [\boldsymbol{l}_1, \boldsymbol{l}_2, \boldsymbol{l}_3]^T \in \mathbb{R}^{3 \times 3}$. Crystal information is traditionally encoded in standardized text format in Crystallographic Information Files (CIFs), which have been the keystone of systematically curated databases of both predicted and experimentally found crystals (Boyd et al., 2019; Zhao et al., 2025) and actively used for materials discovery over the last decades. Fig. 11 exemplifies the structure of a CIF file. CIF databases come with a two-fold challenge: they describe materials' structures and isolated properties but lack holistic information on their functionality, which is present in published papers. Importantly, the textual format of CIFs is less amenable to typical ML pipelines posing barriers to streamlining data-driven materials discovery Tian et al. (2022). While this still remains a grand challenge in materials science, recent literature has increasingly focused on the development of either hand-crafted or machine-learned crystal representations that are well suited for machine learning algorithms.

**Crystal representation learning.** Recent progress in crystal representation learning spans a wide range of representations and modalities, ranging from graph-based to structural and foundation model approaches. CGCNN (Xie & Grossman, 2018) pioneered interpretable crystal graph convolutional networks for property prediction directly from atomic connections, while iCGCNN (Cheng et al., 2021) further enhances this by incorporating Voronoi tessellation and three-body interactions. Physics-guided generative models like PGCGM (Zhao et al., 2023) leverage symmetry-affine transformations to generate diverse, structurally valid crystals, significantly outperforming previous generators. MOFTransformer (Kang et al., 2023) was the first inherently multimodal architecture, combining atom-based and energy-grid embeddings to capture local and global features, achieving SOTA property prediction for MOFs. Similarly, DeepSorption (Cui et al., 2023) integrates global structural awareness via a transformer for highly accurate adsorption predictions in porous materials. More recently, the emergence of foundation models allowed the extension of these practises to broader crystalline systems. MCRT (Feng et al., 2025) multimodally integrates local atomic information with global persistence-image based views of organic molecular crystals, while CLOUD (Xu et al., 2025) employs symmetry-aware, physic-informed string representations for the development of a scalable foundation model, pre-trained on millions of inorganic crystals. Together, these approaches illustrate the shift from local graph-based modeling to multimodal, geometry-aware methodologies, which enable few-shot learning and hold significant promise for leveraging their representations in LLMs.

**LLMs for crystalline materials.** LLMs have recently drawn much interest from the chemistry and materials community due to their unique capabilities in text generation, chemical knowledge integration, and characterization tool utilization (Zheng et al., 2025). Recent efforts have demonstrated the LLMs' capacity to process raw CIF files of inorganic crystals to generate textual descriptions for further language-based training exploitation (Alampara et al., 2025a). Beyond text generation, LLMs have demonstrated promising performance in plausible structure generation, such as CrystaLLM (Antunes et al., 2024), which was trained on millions of inorganic crystal CIF files and validated via ab initio simulations on de-novo generated structures. Chemeleon (Park et al., 2025) proposed the integration of text descriptions with 3D structural data using cross-modal contrastive learning and diffusion models, enabling natural language–guided generation of chemical compositions and structures of inorganic crystals. CSLLM (Song et al., 2025), a framework of three fine-tuned LLMs, use a textual representation for crystal material to predict the synthesizability, synthesis method, and precursors of 3D inorganic crystals. Finally, Matterchat (Tang et al., 2025) is a structure-aware LLM for inorganic materials, trained on more than 140,000 structures, capable of ingesting textual information from atomic structures to reason answers on material description and property-prediction. While these approaches have demonstrated success to small-scale systems, they struggle to generalise to larger, complex systems, such as MOFs, with hundreds or even thousands of atoms per unit cell and latent functionality cues concealed in their CIF representations (Xiao et al., 2023). MOFGPT Badrinarayanan et al. (2025) is the first LLM for de-novo generation of MOFs, utilizing a GPT generator trained on MOFid sequences and a reinforcement learning framework that

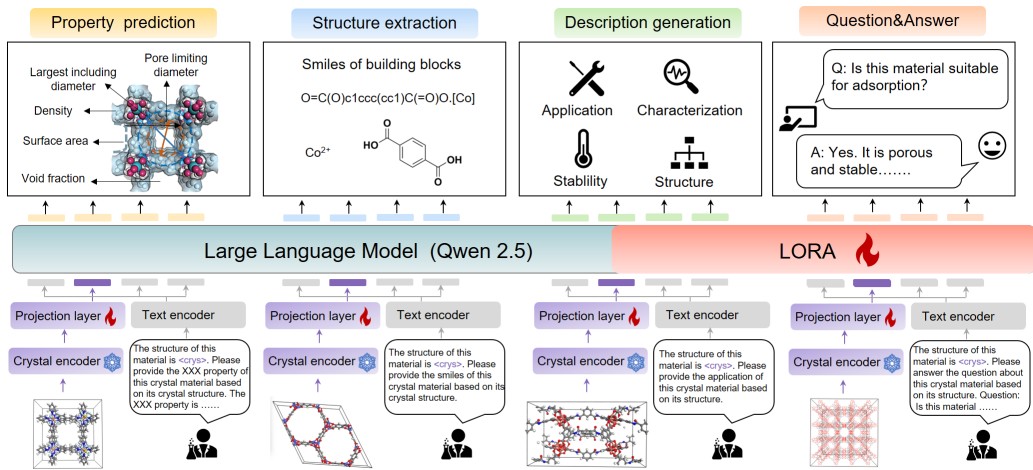

Figure 1: An overview of the L$^2$M$^3$OF framework and its applicability in MOFs design.

steers generation toward target properties—as predicted via MOFormer Kang et al. (2023))—using reward functions. Nevertheless, MOFGPT's generator relies on 2D MOFid strings, missing thus intrinsic functional information encoded in their 3D structure of MOFs. Our work, further complements this expressivity by not only enabling the use of 3D representation of large MOFs structures, but also enhancing the functionality prediction and comprehension of the LLM via jointly training on domain-specific information from the literature.

## 3 DATASET AND MODEL ARCHITECTURE

### 3.1 MOF-SPK DATASET CONSTRUCTION

To train and validate the performance of our model we construct a structure–property–knowledge database for MOFs, namely MOF-SPK. We specifically couple experimentally found MOF structures in CIF format together with curated literature information directly related to these structures. We selected 133,737 CIFs from the Cambridge Crystallographic Data Centre (CCDC) database (Groom et al., 2016) together with the corresponding publications that reported those structures. To ensure data consistency, we removed guest molecules from the MOFs structures using the CSD Python API and applied the StructureMatcher module from the pymatgen Python package (Ong et al., 2013) to identify and eliminate duplicate entries with equivalent crystal structures. We further augmented this data with property calculations and knowledge, leveraging high-throughput computational tools, Python packages and LLMs into a structure–property–knowledge database which formed the test-bench for this study. Based on MOF-SPK we designed four assessment tasks which we then use to evaluate the utility of our LLM as an assistive tool for MOF design. These tasks represent critical and time-consuming characterization procedures in the design of functional materials that can offer valuable acceleration and assistance. The tasks include: i) *property prediction*, ii) *structure extraction*, iii) *description generation*, and iv) *general question & answering*, and form the basis to facilitate model training and evaluation.

Property prediction assesses the LLM's complex physical perception ability on the absolute and relative positions of atoms in the 3D space. Here we computed key physical characteristics of crystal materials — density, pore limiting diameter (PLD), largest cavity diameter (LCD), accessible surface area (ASA), and void fraction (VF) — using high-throughput computational tools such as ZEO++ (Willems et al., 2012). The structure extraction task assesses the LLM's capacity to learn and understand the material's reticular composition of building units, i.e., by extracting the SMILES representation of its molecular building blocks (disjoint fragments of metal and ligands). The MOFid Python toolbox (Bucior et al., 2019) was used to generate the data for this task. For the description generation task, we curated domain knowledge from more than 35,000 scientific publications on experimentally discovered MOFs, extracting information on applications, characterization

methods, stability, and structural features with the assistance of GPT-5-mini and GPT-5-nano. We adopted the method of MOF-ChemUnity for the reliable extraction of MOF information from the literature (Pruyn et al., 2025). Application refers to the uses of the material, such as adsorption or catalysis. Characterization method specifies the experimental techniques that should be employed to analyze the material, for example, Powder X-ray Diffraction or X-ray Photoelectron Spectroscopy. Stability describes how stable the material is, for instance, whether it remains stable in water. Structural features summarize the material's structural characteristics, such as forming a 1D chain or a 3D open framework. The description generation task assesses whether the crystal LLM can learn and establish the relationship between crystal structures and crystal knowledge. Finally, the question & answering task assesses the LLM's ability to answer materials-related questions on MOFs. We generated five questions and answers for each scientific publication based on their abstract. First, we prompted DeepSeek-R1 to identify five keywords from each abstract and then extract relevant questions and answers pairs around those. Examples of the above tasks are included in Section A.3 of the Appendix. To ensure that the MOF-SPK dataset is chemically well-balanced in terms of material representation we perform comprehensive statistical analyses included in Section A.2 of the Appendix.

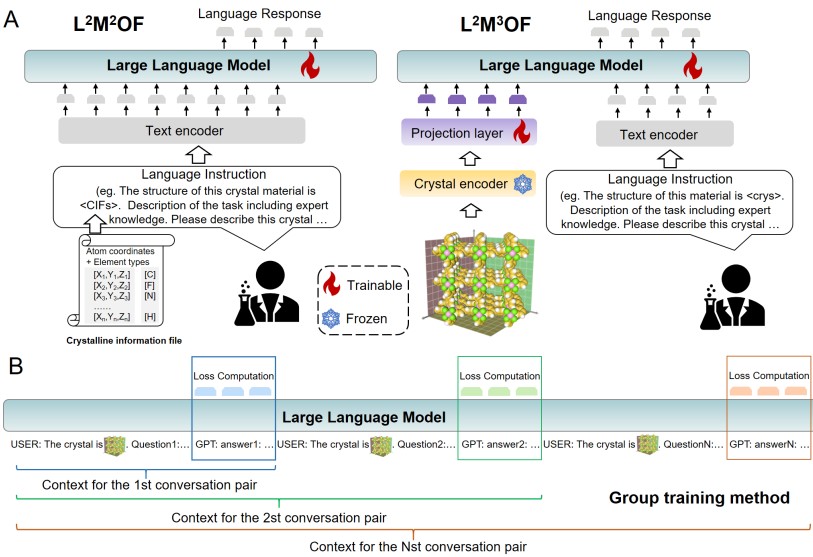

Figure 2: An overview of model architecture and model training methods. (A) The architecture differences between $L^2M^2OF$ and $L^2M^3OF$. (B) The schematic diagram of the group training method.

## 3.2 MODEL ARCHITECTURE

We design two complementary models; apart from our main contribution, $L^2M^3OF$ — a multimodal LLM that incorporates structural information through a MOF 3D structure encoder, we further develop a language-only variant, namely $L^2M^2OF$, which instead represents MOFs in their textual CIF format. The juxtaposition of the two variants against our expertly designed MOF tasks, helps better understand the role of multimodal material representations in LLM-guided discovery. Fig. 2 shows the architectures of the two proposed models.

$L^2M^2OF$ processes a crystal material $\mathcal{M}$ by converting its CIF into a textual sequence $S_{\mathcal{M}}$. This sequence contains the unit cell parameters, space group symmetry, and atomic coordinates with their respective element types. This material representation is concatenated with a task-specific natural language instruction $I_{\mathcal{T}}$ to form the complete input prompt as $X_{\text{LLM}} = [S_{\mathcal{M}}; I_{\mathcal{T}}]$. The instruction $I_{\mathcal{T}}$ embeds expert domain knowledge to guide the model. For instance, for VF prediction, $I_{\mathcal{T}}$ defines the property and its physical significance. The model then generates the target prediction $Y$

autoregressively. The probability of generating the output sequence of $L$ tokens is given by:

$$P(Y \mid X_{\text{LLM}}) = \prod_{i=1}^{L} P(y_i \mid y_{<i}, X_{\text{LLM}}; \Theta_{\text{LLM}}), \tag{1}$$

where $\Theta_{\text{LLM}}$ represents the parameters of a pre-trained LLM. A key advantage of this text-only paradigm is its ability to perform inference with SOTA commercial LLMs (e.g., GPT-5, Gemini-Pro, Deepseek-R1) without additional training. In this study, we also fine-tuned an open-source LLM on the MOF-SPK database to serve as a strong text-only baseline for comparison against our multimodal approach.

$L^2M^3OF$ conversely fuses textual instructions with non-textual, geometric structural data. The model consists of three core components:

**Crystal Structure Encoder:** We employ PMTransformer (Park et al., 2023) as the crystal encoder, which is a GNN pre-trained on 1.9 million hypothetical porous materials. It takes the crystal structure $\mathcal{M}$ and outputs a fixed-dimensional latent representation, or embedding, $\mathbf{z}_{\text{struct}} \in \mathbb{R}^d$:

$$\mathbf{z}_{\text{struct}} = \text{PMTransformer}(\mathcal{M}; \Theta_{\text{PMT}}), \tag{2}$$

where $\Theta_{\text{PMT}}$ represents the frozen, pre-trained parameters of the encoder.

**Multimodal Projection Bridge:** This component transforms and compresses the structural embedding for seamless integration with the language model through a compression and projection network. The compression network, $\text{MLP}_{\text{token}}$, then compresses this sequence along the token dimension from length $N$ to a shorter, fixed length $M$ ($M < N$). We empirically found that this compression accelerates training significantly without loss of performance. The projection network, $\text{MLP}_{\text{feat}}$, projects the encoder's output from its native dimension $d_{\text{enc}}$ to a sequence of $N$ tokens in LLMs' embedding space $\mathbb{R}^{d_{\text{LLM}}}$. The entire process is defined as:

$$\mathbf{H}_{\text{struct}} = \text{MLP}_{\text{token}}(\mathbf{z}_{\text{struct}}; \Theta_{\text{token}}), \quad \mathbf{H}_{\text{proj}} = \text{MLP}_{\text{feat}}(\mathbf{H}_{\text{struct}}; \Theta_{\text{feat}}), \tag{3}$$

where $\Theta_{\text{bridge}} = (\Theta_{\text{feat}}, \Theta_{\text{token}})$ denotes the combined parameters of the projection and compression MLPs, and $\mathbf{H}_{\text{struct}} \in \mathbb{R}^{M \times d_{\text{enc}}}$, $\mathbf{H}_{\text{proj}} \in \mathbb{R}^{M \times d_{\text{LLM}}}$.

**Large Language Model:** The compressed structural token sequence $\mathbf{H}_{\text{struct}}$ is prepended to the tokenized instruction sequence $\text{Tokenize}(I_{\mathcal{T}})$ to form the combined input for the LLM. The LLM then generates the output conditioned on this multimodal input:

$$P(Y \mid \mathcal{M}, I_{\mathcal{T}}) = \prod_{i=1}^{L} P(y_i \mid y_{<i}, \mathbf{H}_{\text{struct}}, I_{\mathcal{T}}; \Theta_{\text{LLM}}, \Theta_{\text{bridge}}). \tag{4}$$

During training, the encoder parameters $\Theta_{\text{PMT}}$ are kept frozen to preserve its pre-trained knowledge and stabilize training. Only $\Theta_{\text{Bridge}}$ and $\Theta_{\text{LLM}}$ are updated.

## 3.3 TRAINING OBJECTIVE

We trained our models using an instruction-tuning paradigm, tailoring them for property prediction tasks in materials science. The objective is to minimize the negative log-likelihood of the target sequence (e.g., the numerical or classification values) given the input instruction and material data.

For a dataset $\mathcal{D}$ of $N$ examples, each containing an instruction, a material, and a target response $(I_{\mathcal{T}}^{(i)}, \mathcal{M}^{(i)}, Y^{(i)})i = 1^N$, the loss function $\mathcal{L}$ for L$^2$M$^3$OF is defined as:

$$\mathcal{L}(\Theta_{\text{LLM}}, \Theta_{\text{bridge}}) = -\frac{1}{N} \sum_{i=1}^{N} \log P(Y^{(i)} \mid \mathcal{M}^{(i)}, I_{\mathcal{T}}^{(i)}; \Theta_{\text{LLM}}, \Theta_{\text{bridge}}). \tag{5}$$

This supervised fine-tuning (SFT) process teaches the model to follow instructions and reason about material properties based on the provided textual and structural information, enabling it to generalize to new, unseen materials and tasks. We further implement a *group training strategy* to enhance context diversity during SFT. For each mini-batch, instruction–answer pairs are first sampled according

to the standard batching procedure, and then multiple pairs within the same batch are randomly grouped and concatenated to form multi-turn conversational samples. The loss is computed only on the answer tokens of each question within the group, while the preceding pairs serve as contextual background. This approach effectively increases the diversity of training contexts without substantially increasing computational costs, since grouping instruction-answer pairs within the batch changes only how the samples are combined and the actual batch size, but not the total number of tokens within the batch. The method acts as an efficient form of data augmentation, exposing the model to richer contextual patterns during training.

# 4 EXPERIMENTS

## 4.1 EXPERIMENTAL SETUP

**Backbones and adaptation.** For structure encoding we use PMTransformer (frozen) and for the language backbone we use Qwen2.5-7B-Instruct for both $L^2M^2OF$ (text-only) and $L^2M^3OF$ (multimodal). In $L^2M^3OF$, the PMTransformer output is passed through the projection bridge (Section 3.2) compresses them into $M=16$ structural tokens that are embedded in to the instruction tokens. The LLM is adapted with LoRA with rank $r=8$, $\alpha=16$, and dropout 0.05. $L^2M^2OF$ uses the same Qwen2.5 backbone and LoRA settings but receives CIF text as its material representation (no structural tokens).

**Training setup.** We use AdamW ($\beta_1=0.9$, $\beta_2=0.999$, weight decay 0), cosine LR schedule with peak LR $2\times10^{-4}$ and warmup ratio 0.03, bf16 with activation checkpointing, and an effective batch size of 256 Q&A pairs via gradient accumulation. $L^2M^2OF$ and $L^2M^3OF$ was trained on $8\times$ and $4\times$H100, respectively. $L^2M^2OF$ needs more GPU memory for the same batch size because the CIF takes a lot more tokens than compressed structural embedding tokens. We train the models for 2,000 steps unless specified. The fine-tuning GPU hours of $L^2M^3OF$ is 25.87 and the fine-tuning GPU hours of $L^2M^2OF$ is 551.29.

## 4.2 EXPERIMENTAL RESULTS

Training models on past data and evaluating them on future discoveries is crucial because it mirrors real-world deployment scenarios. To this end, we partition the dataset by material deposition year, i.e., crystal structures deposited on or before 2020 were used for training, while those from 2021 onwards formed the validation set. We further sample 500 crystal structures deposited after 2022 and use as the test set. We evaluate the performance of our models against leading commercial LLMs

Table 2: Performance comparison of commercial LLMs, $L^2M^2OF$, and $L^2M^3OF$ on property prediction and structure extraction. The best performances are in **bold**, the second best underlined.

| Metric | DeepSeek-V3 | DeepSeek-R1 | GPT-4o | GPT-5 mini | GPT-5 | Gemini-2.5-pro | $L^2M^2OF$ | $L^2M^3OF$ |
|---|---|---|---|---|---|---|---|---|
| **Property Prediction (MAE)** | | | | | | | | |
| PLD ($\downarrow$) | 1.99 | 1.97 | 2.94 | 2.93 | 3.24 | 2.09 | 1.19 | **0.49** |
| LCD ($\downarrow$) | 2.27 | 3.10 | 4.14 | 4.59 | 4.37 | 2.28 | 1.04 | **0.47** |
| Density ($\downarrow$) | 0.41 | 0.35 | 9.86 | 0.31 | 0.31 | 0.31 | 0.20 | **0.19** |
| ASA ($\downarrow$) | 762.7 | 1481.6 | 745.3 | 1317.6 | 726.2 | 805.9 | 492.6 | **188.7** |
| VF ($\downarrow$) | 0.21 | 0.39 | 0.13 | 9.63 | 0.08 | 0.88 | 0.04 | **0.01** |
| **Structure Extraction** | | | | | | | | |
| BLEU ($\uparrow$) | 0.27 | 0.28 | 0.20 | 0.38 | 0.38 | 0.38 | **0.45** | 0.31 |
| EXACT ($\uparrow$) | 0.00 | 0.01 | 0.00 | 0.02 | 0.02 | 0.00 | **0.25** | 0.16 |
| MACCS ($\uparrow$) | 0.50 | 0.52 | 0.49 | 0.53 | 0.56 | 0.57 | **0.68** | 0.48 |
| RDK ($\uparrow$) | 0.32 | 0.40 | 0.27 | 0.37 | 0.43 | 0.46 | **0.48** | 0.22 |
| MORGAN ($\uparrow$) | 0.22 | 0.25 | 0.18 | 0.24 | 0.28 | 0.29 | **0.40** | 0.20 |
| VALIDITY ($\uparrow$) | 0.34 | 0.44 | 0.35 | 0.44 | 0.60 | 0.80 | 0.71 | **0.90** |

by Google (Comanici et al., 2025), DeepSeek Guo et al. (2025) and OpenAI (OpenAI et al., 2024) on the four tasks introduced in Section 3.1 to assess the learning and comprehensive capabilities of LLMs for MOFs. Here we do not compare against other crystal LLMs from the literature as these are not suitable for MOF materials or do not support knowledge generation capabilities which is one of the main scopes of this study.

**Property prediction.** This task assesses the LLMs' performance (in terms of mean absolute error) in accurately predicting a wide range of MOF properties. While the main aim of this work is rather to facilitate MOF representation and knowledge understanding, property prediction performance on geometry-induced properties such as for example PLD, LCD, and ASA, can correlate to the model's understanding on the broader physical 3D structures of MOFs, linking to higher-level conceptualisation tasks. As shown in the top section of Table 2 and Fig. 12, $L^2M^3OF$ attains the lowest MAE on all five targets among all the large language models, while commercial LLMs consistently underperform, especially on geometry-sensitive metrics such as PLD, LCD, and ASA, which require robust grounding in 3D pore topology. Even on the easier task of density prediction, the best commercial systems (Gemini-2.5-Pro, GPT-5, GPT-5-mini) still perform worse against $L^2M^3OF$. Importantly, we further observe clear failure modes suggestive of hallucination or unit/normalization errors: GPT-4o reaches MAE = 9.86 on density and GPT-5 mini reaches MAE = 9.63 on void fraction. The $L^2M^2OF$ variant also outperforms the commercial LLMs on property prediction, albeit 'losing' against the multimodal $L^2M^3OF$. Under same number of training steps however, $L^2M^2OF$ is substantially slower because textual crystal descriptions require far more tokens. These findings demonstrate the utility of literature injected domain knowledge in the training of scientific LLMs and further indicate that multimodal training enhances an LLM's ability to perceive and reason about the 3D spatial information of porous crystalline materials significantly, yielding superior accuracy as well as efficiency. Table 4 in Appendix A.4 presents additional results on head-to-head property prediction comparisons between our proposed language models against leading MOF-specialised models, namely CGCNN Xie & Grossman (2018) and MOFTransformer Kang et al. (2023). Results substantiate $L^2M^3OF$'s competitive performance even against the state-of-the-art MOFTransformer.

**Structure extraction.** Extracting molecular building blocks from MOFs requires a fine-grained perception of local chemical information and structural features. Here we compare the SMILES of LLM-extracted units against ground-truth (as computed in Section 3.1 to assess accuracy and validity according to the BLEU, EXACT and VALIDITY normalized scores as in (Zhuang et al., 2025)[1]. We further assess structural similarity between the extracted molecular units and the ground-truth. We test three different molecular fingerprinting methods, namely MACCS, RDKit and Morgan and use the Tanimoto similarity metric (Szafarczyk et al., 2024). Interestingly, LLMs that use CIFs as textual input achieve even stronger results on this task. In particular, $L^2M^2OF$ performs best on BLEU, EXACT, MACCS, RDK, and MORGAN, while Gemini-2.5-pro demonstrated the second-best performance in the on BLEU, MACCS, RDK, and MORGAN, with a high SMILES VALIDITY score of 0.8, which highlights the strong capabilities of advanced commercially available LLMs in chemical tasks. The success of CIF-based models is not entirely surprising however; the explicit textual representation of CIF directly encodes the elemental composition of materials, which facilitates the inference of constituent molecules and metallic units putting more emphasis on local environment. In contrast, vectorized crystal representations make such local compositional information less transparent.

**Description generation.** The description generation task is evaluated across four subtasks: application recommendation, characterization method, stability description, and structural feature. Among these, application recommendation is both the most important and the most challenging, as it requires not only an accurate perception of crystal structures and properties but also sufficient domain knowledge to map materials to plausible use cases. This is an indispensable tool to support scientists in making informed, application-oriented decisions rather than treating structure analysis in isolation. To ensure reliable evaluation and reduce reliance on manual judgment, we employ multiple commercial LLMs, including o4-mini, GPT-5, DeepSeek-reasoner, and DeepSeek-chat as impartial chemistry knowledge 'referees'. Following (Wang et al., 2023), we adopt a calibration strategy where commercial LLMs compares the outputs of two LLMs for each test question. Since LLMs are sensitive to response order, we mitigate positional bias by swapping the order of the outputs and re-evaluating. The final score is computed by aggregating results from both prompt orders. Each evaluation includes the question, a ground-truth answer and two candidate responses. During assessment, the referee LLMs are instructed to select best LLM responses based on strict scientific accuracy and factual correctness. Fig. 10 in Appendix A.3 illustrates an example prompt for descrip-

---

[1]The SMILES BLEU score measures the overlap between the LLM-generated and ground-truth SMILES strings. EXACT assesses exact SMILES matches. VALIDITY evaluates the percentage of the LLM-generated molecules that conform to chemical syntax rules (Zhuang et al., 2025).

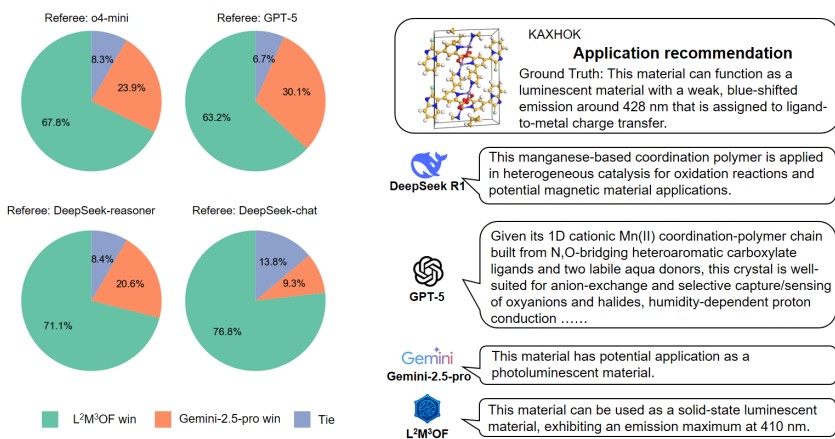

Figure 3: Performance comparison of Gemini-2.5-pro and $L^2M^3OF$ on the tasks of description generation and a case study of application recommendation task on MOF KAXHOK.

tion evaluation. During the evaluation process, the temperature of commercial LLMs is the default value (temperature=1).

Given its strong preliminary performance on property prediction and structure extraction, we adopt Gemini-2.5-Pro as a baseline for comparisons. As shown in in the left-most side of Fig. 3, $L^2M^3OF$ outperforms Gemini-2.5-Pro on 1,589 description generation tasks, across all LLM referees. Conducting a sign test at a 95% confidence level reveals that $L^2M^3OF$ consistently outperforms Gemini-2.5-Pro with a $p$-value of 0.00004 with Bonferroni correction. Qualitatively, our model not only provides applications that are more closely aligned with the ground truth, but also offers more detailed information as illustrated in the example of Fig. 3 (right-most side).

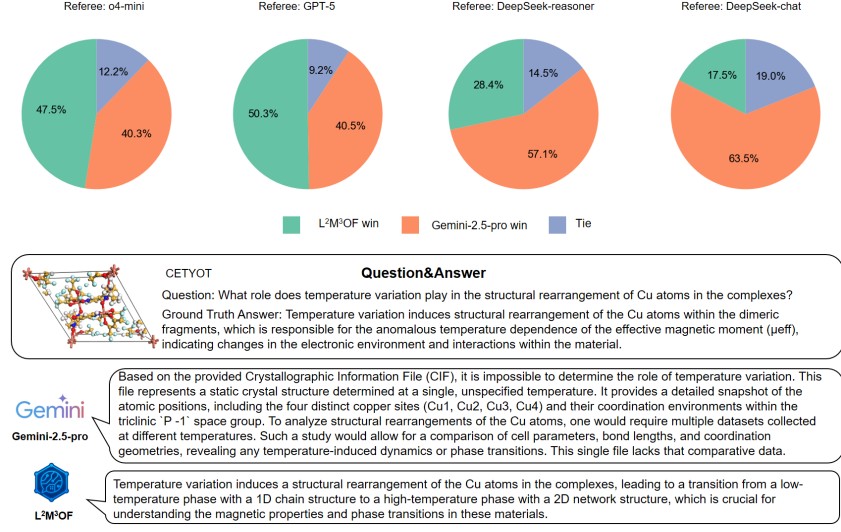

Figure 4: Performance comparison of Gemini-2.5-pro and $L^2M^3OF$ in the tasks of question&answer.

**Question & Answering.** The question and answering task not only relies on understanding the crystal structure but also emphasizes the model's mastery of MOF-specific domain knowledge. $L^2M^3OF$ demonstrated very competitive performance against Gemini-2.5-pro on this task, yet with no statistical significance. An assessment disparity between Deepseek and OpenAI referess was also observed, with the former deeming Gemini-generated answers more accurate. Qualitatively, we generally observed that Gemini-2.5-pro often produced overly verbose responses and failed to

reason toward the correct answer due to its lack of domain knowledge, whereas $L^2M^3OF$ was able to reason and respond concisely and more accurately.

**Downstream tasks.** In this paragraph we test the performance difference between our two proposed language models, $L^2M^3OF$ and $L^2M^2OF$. We select a wider range of 15 functional property downstream tasks, inline with Kang et al. (2023), including band gap, gas adsorption and stability, amongst others, to assess their broader predictive performance. Among those 15 downstream tasks, $L^2M^3OF$ outperformed $L^2M^2OF$ in 14 tasks, especially in the gas adsorption task which is more dependent on material spatial perception ability (Fig 14, Appendix A.4). However, $L^2M^2OF$ only outperformed $L^2M^3OF$ in the band gap prediction task that mainly relies on local element information. From detailed comparisons, $L^2M^3OF$ emerges as a competitive universal property predictor, besides a meticulous chemistry-aware knowledge generator, further amplifying its utility as an indispensable tool for materials practitioners.

Table 3: Ablation studies performance comparison.

| | Property Prediction | | | | | Structure Extraction | | | | | |
|---|---|---|---|---|---|---|---|---|---|---|---|
| | PLD ($\downarrow$) | LCD ($\downarrow$) | Density ($\downarrow$) | ASA($\downarrow$) | VF ($\downarrow$) | BLEU ($\uparrow$) | EXACT ($\uparrow$) | MACCS ($\uparrow$) | RDK ($\uparrow$) | MORGAN($\uparrow$) | VALIDITY($\uparrow$) |
| **$L^2M^3OF$** | 0.49 | 0.47 | 0.19 | 188.74 | 0.01 | 0.31 | 0.16 | 0.48 | 0.22 | 0.20 | 0.90 |
| w/o joint training | 0.67 | 0.69 | 0.36 | 348.49 | 0.01 | 0.19 | 0.01 | 0.32 | 0.14 | 0.11 | 0.63 |
| w/o group training | 0.86 | 0.90 | 0.26 | 291.58 | 0.02 | 0.27 | 0.21 | 0.42 | 0.18 | 0.17 | 0.90 |
| w/o supervised fine-tuning | 0.70 | 1.14 | 0.24 | 1826.16 | 0.02 | 0.15 | 0.04 | 0.27 | 0.09 | 0.10 | 0.54 |

## 4.3 ABLATION STUDIES

To probe cross-task interactions, we compare joint training and separate training across different tasks using the same data budget and model size (Table 3 and Fig 13). The results show clear, consistent gains from joint training. On property prediction, the jointly trained model achieves substantially lower MAE on geometry-dependent targets. There is also a significant improvement in the structure extraction task. In the description generation task, the head-to-head win rate against Gemini-2.5-Pro rises from 59.0% to 67.8% via using o4-mini as referee. The three tasks capture complementary facets of the same underlying material representation. Property prediction forces the model to be numerically faithful to pore geometry; structure extraction sharpens the model's awareness of local chemistry; description generation ties these cues to functional outcomes. Optimizing them together encourages a holistic, structure-aware embedding that captures both global topology and local chemical context. We also investigate the effect of group training which, without introducing additional training overhead, primarily improves the predictive accuracy of the model on the property prediction task. The above experimental results further demonstrate the importance of enabling the model to jointly learn and capture crystal structures, properties, and knowledge.

To isolate the contribution of supervised fine-tuning on the LLM and assess the benefits of multimodal alignment alone, we also experimented with training the model while keeping the LLM frozen (Table 3 and Fig 13). The experimental results show that although the model without SFT significantly underperforms against the $L^2M^3OF$ baseline, it still outperforms Gemini-2.5-pro in the tasks of property and description generation. This step demonstrates the significant importance of multimodal alignment for large language models to enable materials structure understanding, and especially spatial structure information. We also investigate performance sensitivity to projection size (M tokens). Table 5 in Appendix A.4 reveals an insignificant performance gain as projection size increases, coupled however with slower training times.

## 5 CONCLUSIONS

We proposed $L^2M^3OF$, the first multimodal large language model designed specifically for MOFs. By integrating geometric structure encoding with language-based domain knowledge, $L^2M^3OF$ outperforms state-of-the-art commercial LLMs across property prediction, description generation, and question answering tasks – despite using fewer parameters. These results highlight the importance of multimodal architectures in capturing the intricate interplay between structure and function in crystalline materials. $L^2M^3OF$'s success demonstrates how grounding LLMs in 3D representations and curated literature can bridge gaps in automated materials discovery. As a lightweight and versatile tool, it offers chemists a scalable AI assistant for navigating complex design spaces.

## 6 REPRODUCIBILITY STATEMENT

To ensure our findings are reproducible, we'll make all code and processed data publicly available upon paper acceptance. The dataset construction is detailed in Section 3.1, and we will share the processed data to facilitate its use by others. The model architecture is fully described in Section 3.2, and training specifics are provided in Section 3.3. The evaluation protocols are laid out in detail in the Section 4.2. We commit to making both our training and inference code accessible, allowing for full replication of our experiments. This comprehensive approach ensures that our results can be validated and built upon by the research community.

## 7 ETHICS STATEMENT

While our training data is confined to scientific literature on MOFs, the underlying base model carries potential societal biases and inherent safety risks. Consequently, our model's outputs do not ensure 100% accuracy and may not comprehensively cover the full spectrum of safety and honesty. The model should, therefore, only be used under professional guidance to prevent the generation of biased, inaccurate, or harmful content in real-world applications.

To mitigate these risks and ensure responsible future development, we recommend a series of safeguards. The deployment of this model in real-world scientific research should be accompanied by professional guidance. Future research should prioritize a more comprehensive evaluation of the base model's safety and explore integrating formal safety and honesty constraints directly into the trained $L^2M^3OF$ architecture. These crucial steps can help ensure that further advancements in this field are built upon a strong foundation of ethical responsibility.

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

# A   APPENDIX

## A.1   THE USE OF LARGE LANGUAGE MODELS

In this study, large language models were employed in several aspects of our work. During manuscript preparation, we used LLMs for polishing the language. In the research process, LLMs were applied to literature corpus processing, benchmark testing on the MOF-SPK database, and serving as evaluators of experimental results. The specific usage details and the models adopted are provided in the main text. All intellectual contributions such as research ideas, experimental designs, analyses, and conclusions were developed solely by the authors, who take full responsibility for the content of this paper.

## A.2   MOF-SPK STATISTICAL ANALYSIS

In this section we examine the underlying chemical balance of the MOF-SPK dataset in its three main aspects, structure, property and knowledge. In terms of elemental diversity, MOF-SPK is quite diverse: excluding noble gases that do not form coordination bonds, the database contains up to 81 chemical elements (Fig. 5A). We further examine the distribution of MOF sizes in the database, expressed in number of LLM tokens, to assess the representational bias of diverse materials when processed by LLMs. For this we used the Qwen2.5-7B (Yang et al., 2025) tokenizer to quantify the token-length distribution of CIF representations for MOFs. The dataset exhibits a unimodal distribution with a peak between $10^3$ and $10^4$ tokens, indicating that textual serialization of crystal structures typically requires thousands to tens of thousands of tokens (Fig. 5B). Notably, the most verbose case, the CIF of material LELMEW, reaches 94,000 tokens while the structure contains only 3216 atoms, underscoring the substantial redundancy introduced by purely text-based encodings of complex crystalline materials. These observations motivate more compact representations, such as structured symbols or multimodal embeddings, that capture geometric and compositional information without incurring excessive sequence length. We further analyzed the distributions of five key MOF properties, namely LCD, PLD, density, ASA, and VF, crucial for understanding MOFs' physical characteristics. Their distributions exhibit long-tailed behavior, highlighting the inherent challenges in predicting these properties (Fig. 5C). We use Qwen2.5-7B to analyze and summarize the application landscape of MOFs. The results show substantial breadth: beyond uses such as gas adsorption and separation, catalysis, and chemical sensing, MOFs also function as luminescent materials and as crystalline sponges for host–guest chemistry (Fig. 5D). This diversity creates an opportunity for LLMs to exceed the capabilities of individual human experts, who are typically specialized in a single application area and may overlook cross-domain potential. For example, a

framework designed by an adsorption specialist might underperform on uptake targets yet be an excellent catalyst; domain boundaries can mask such good alternative uses.

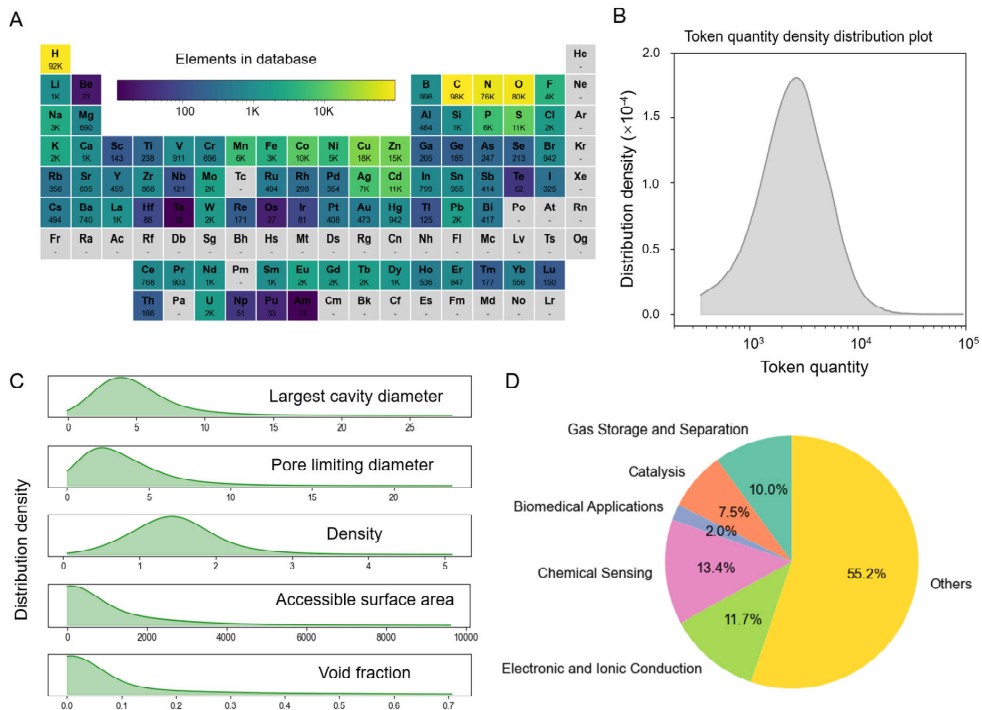

Figure 5: Data analysis of structure–property–knowledge database for crystal materials. (A) Elemental distribution in the dataset. (B) The token quantity density distribution of the CIFs in the dataset. (C) The distribution of properties of crystal material in the dataset. (D) The application distribution of the crystal material in the dataset.

### A.3 DATASET EXAMPLE

**Prompt**

The structure of the crystal material is `<crys>`. Can you provide the pore limiting diameter of this crystal material according to the structure of this crystal material? The pore limiting diameter (PLD) in crystal materials refers to the smallest diameter of a sphere that can pass through the pore structure of the material. This diameter represents the smallest void space in the pore system, and it's crucial for understanding the material's size-selective capabilities, especially in applications like gas separation. The output format is [[pore limiting diameter]] and the unit of the output is Å. The probe molecule is the Nitrogen molecule, and the radius of the probe molecule is 1.82 Å.

**Ground-truth**

[[1.5 Å]]

Figure 6: Example of prompt and ground truth for property prediction task.

### A.4 ADDITIONAL RESULTS

---

**Prompt**

The structure of the crystal material is `<crys>`.The Simplified Molecular Input Line Entry System (smiles) is a specification in the form of a line notation for describing the structure of chemical species using short ASCII strings. SMILES strings can be imported by most molecule editors for conversion back into two-dimensional drawings or three-dimensional models of the molecules.Can you provide the smiles of this crystal material according to the structure of this crystal material.The format of the output is {{SMILES}}.

---

**Ground-truth**

{{O=C1N=NC(=O)c2cnccc21.[Zn]}}

---

Figure 7: Example of prompt and ground truth for structure extraction task.

---

**Prompt**

The structure of the crystal material is `<crys>`. Please provide the Application of this crystal material in 100 words according to the structure of this crystal material.

---

**Ground-truth**

Anti-corrosion coatings for electrogalvanized steel
The reason is: The study demonstrates that binary zinc carboxylate MOFs (e.g., $ZnC_{12}C_{16}$) form compact, hydrophobic layers on metal surfaces. These coatings act as barriers against corrosion by inhibiting zinc dissolution and reducing solubility. Their performance surpasses single carboxylates (e.g., $Zn(C_{12})_2$ or $Zn(C_{16})_2$) in electrochemical and climatic tests, aligning with industrial requirements for eco-friendly alternatives to phosphating/chromating processes.

---

Figure 8: Example of prompt and ground truth for description generation task.

---

**Prompt**

The structure of the crystal material is `<crys>`. Please answer this question in 100 words according to the structure of this crystal material. Question: What advantages do the new zinc-based metal-organic frameworks (MOFs) offer in terms of environmental impact and synthesis?

---

**Ground-truth**

The new zinc-based MOFs can be easily synthesized in water, providing an environmentally friendly anti-corrosion treatment for metals, which aligns with sustainable practices in material science.

---

Figure 9: Example of prompt and ground truth for question&answer task.

Table 4: Performance comparison of CGCNN, MOFTransformer, $L^2M^2OF$, and $L^2M^3OF$ on property prediction. Best performances are boldfaced while second-best underscored.

| Metric | CGCNN | MOFTransformer | $L^2M^2OF$ | $L^2M^3OF$ |
|---|---|---|---|---|
| **Property Prediction (MAE)** | | | | |
| PLD ($\downarrow$) | 1.17 | **0.39** | 1.19 | 0.49 |
| LCD ($\downarrow$) | 1.30 | **0.37** | 1.04 | 0.47 |
| Density ($\downarrow$) | 0.14 | **0.11** | 0.20 | 0.19 |
| ASA ($\downarrow$) | 412.4 | 482.8 | 492.6 | **188.7** |
| VF ($\downarrow$) | 0.03 | **0.01** | 0.04 | **0.01** |

---

**Prompt**

You are an expert in crystal materials. Please evaluate which of the two given answers is better based on the question and the standard answer. Please base your judgments on scientific evidence regarding their proximity to the standard answer, rather than on other non-scientific factors.

The judgment is mainly based on the following several criteria:

1.The correctness of the answer. Make a judgment based on the closeness of the answer to the standard answer.

2.The accuracy and detail of the answer. Broad and general answers are not acceptable.

Please only output the serial number of the better answer.

The question is: {Question}

The standard answer is: {Standard answer}

The given answer 1 is: {Answer 1}

The given answer 2 is: {Answer 2}

Figure 10: Example of prompt for evaluation.

Figure 11: Example of crystallographic information file.

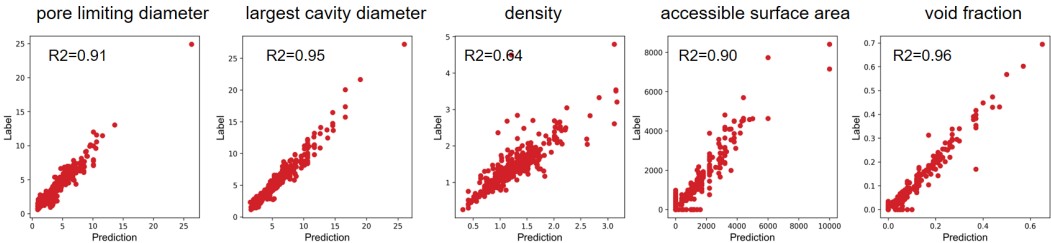

Figure 12: Parity plots and $R^2$ performance of $L^2M^3OF$ across the various property prediction tasks.

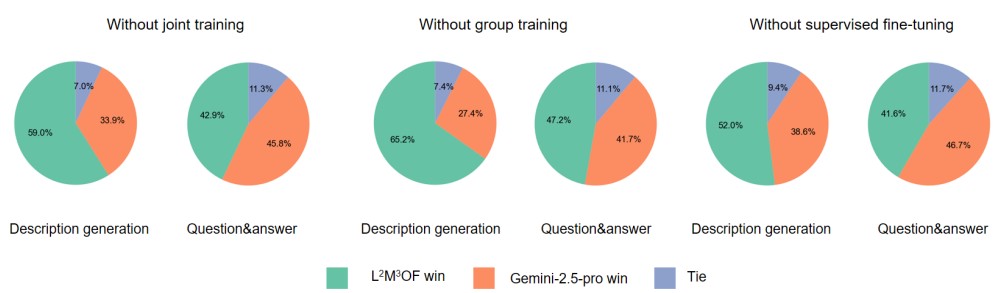

Figure 13: Performance comparison of Gemini-2.5-pro and $L^2M^3OF$ in the ablation study.

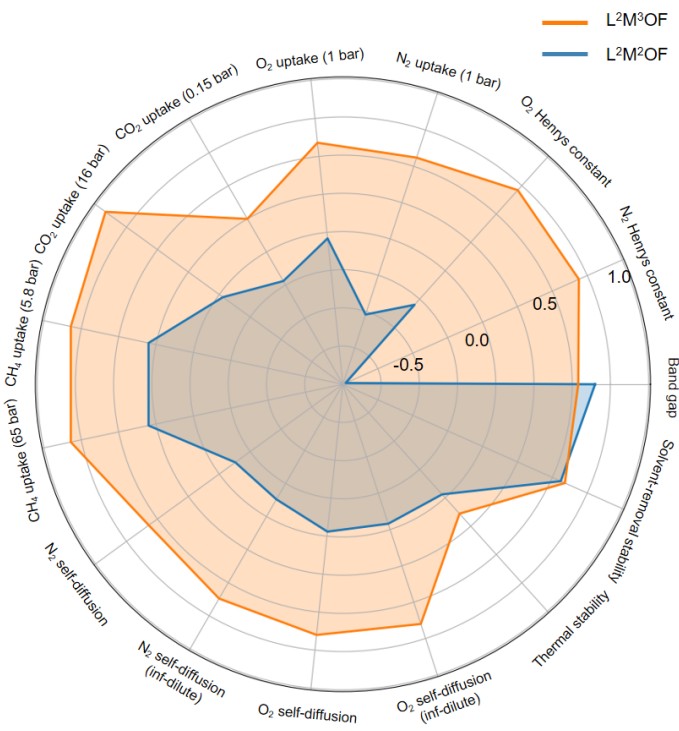

Figure 14: Performance comparison of $L^2M^3OF$ and $L^2M^2OF$ on 15 MOF property prediction downstream tasks. The radial plot reports accuracy for the solvent-removal stability task and $R^2$ for the rest of the tasks.

Table 5: Performance comparison of L$^2$M$^3$OF with different M tokens projection size.

| Metric | M tokens=1 | M tokens=16 | M tokens=32 | M tokens=64 | M tokens=128 | M tokens=128 |
|---|---|---|---|---|---|---|
| **Property Prediction (MAE)** | | | | | | |
| PLD ($\downarrow$) | 0.49 | 0.49 | 0.51 | 0.59 | 0.65 | 0.58 |
| LCD ($\downarrow$) | 0.51 | 0.47 | 0.48 | 0.53 | 0.61 | 0.55 |
| Density ($\downarrow$) | 0.18 | 0.19 | 0.16 | 0.18 | 0.19 | 0.18 |
| ASA ($\downarrow$) | 195.2 | 188.7 | 180.2 | 204.7 | 221.4 | 224.7 |
| VF ($\downarrow$) | 0.01 | 0.01 | 0.01 | 0.01 | 0.01 | 0.01 |
| **Structure Extraction** | | | | | | |
| BLEU ($\uparrow$) | 0.29 | 0.31 | 0.31 | 0.29 | 0.28 | 0.28 |
| EXACT ($\uparrow$) | 0.16 | 0.16 | 0.19 | 0.13 | 0.17 | 0.14 |
| MACCS ($\uparrow$) | 0.44 | 0.48 | 0.50 | 0.44 | 0.44 | 0.43 |
| RDK ($\uparrow$) | 0.19 | 0.22 | 0.25 | 0.20 | 0.21 | 0.17 |
| MORGAN ($\uparrow$) | 0.18 | 0.20 | 0.23 | 0.19 | 0.19 | 0.18 |
| VALIDITY ($\uparrow$) | 0.83 | 0.90 | 0.85 | 0.83 | 0.90 | 0.86 |
| **Cost evaluation** | | | | | | |
| GPU hours ($\downarrow$) | 23.72 | 25.87 | 27.99 | 32.56 | 42.03 | 62.01 |

