# OpenReview forum: "L^2M^3OF: A Large Language Multimodal Model for Metal-Organic Frameworks"
_ICLR.cc/2026/Conference — Submitted to ICLR 2026_

### Official Review · Reviewer_H4S1 · 2025-10-22

**Soundness:** 3
**Presentation:** 3
**Contribution:** 3
**Rating:** 8
**Confidence:** 4

**Summary:**

The paper proposes $L^2M^3OF$, the first multimodal LLM for MOFs, which integrates crystal representation learning with language understanding to process structural, textual, and knowledge modalities jointly. The model uses a pre-trained crystal encoder with a lightweight projection layer to efficiently align with language instructions. Benchmarked on a structure–property–knowledge database of crystalline materials curated by the authors, $L^2M^3OF$ outperforms leading text-based closed-source LLMs in property prediction and
knowledge generation tasks, despite using far fewer parameters. Such results highlight the importance of multimodal approaches for porous crystalline material understanding.

**Strengths:**

1. The authors curated a structure–property–knowledge database for MOFs, namely MOF-SPK, and consequently designed structured subtasks that collectively cover both quantitative and qualitative reasoning about materials, thus providing a comprehensive benchmark for evaluating the capability of LLMs for porous crystalline material understanding.

1. A two-stage MLP bridge maps these latent vectors into the token space of Qwen-2.5, enabling seamless cross-modal attention.

1. The authors freeze PMTransformer weights to retain structural priors while fine-tuning only the bridge and LLM, avoiding catastrophic forgetting. They further propose “group training”, a lightweight conversational data-augmentation scheme that concatenates multiple Q&A pairs in one batch, improving contextual diversity without increasing token count.

**Weaknesses:**

1. The evaluation uses GPT-4-mini as an automatic judge in description generation and Q&A tasks. While this provides consistency, it may introduce model bias—the evaluation may favor certain phrasing styles or reasoning formats aligned with GPT-family models. Human expert evaluation or hybrid methods would strengthen validity.

1. In description generation and Q&A tasks, the performance gap versus Gemini-2.5-Pro is small, which suggests that the overall advantage of $L^2M^3OF$ remains modest compared to large-scale commercial models.

1. The paper does not report inference or fine-tuning cost. The $L^2M^3OF$’s multimodal architecture (PMTransformer + bridge + LLM) likely incurs higher computational overhead than text-only baselines (like $L^2M^2OF$).

**Questions:**

1. Have the authors experimented with training the model with LLM frozen? I suppose this would isolate the contribution of SFT on the LLM, and also tells us how much gain comes from multimodal alignment alone vs. from updating the language model. Especially, the work is compared with baselines of closed-source LLMs whose parameters can't be fine-tuned.

1. There are multiple existing works that train crystal GNNs on MOF data. Therefore, is there any specific reason to justify the choice of PMTransformer as the structure encoder? Also, how large is the impact of the model architecture choice on the performance of $L^2M^3OF$?

---

> ### Author Response · Authors · 2025-12-03
> **Point-by-point replies to the comments1**
>
> We thank the reviewer for recognizing our contributions in introducing the first multimodal LLM for MOFs and demonstrating its advantages through efficient cross-modal alignment and a comprehensive MOF structure–property–knowledge benchmark. Point-by-point replies to the comments of reviewer are detailed as follows:
>
> 1. The evaluation uses GPT-4-mini as an automatic judge in description generation and Q&A tasks. While this provides consistency, it may introduce model bias—the evaluation may favor certain phrasing styles or reasoning formats aligned with GPT-family models. Human expert evaluation or hybrid methods would strengthen validity.
>
> We would like to thank the reviewer for pointing this potential bias issue out. Evaluating responses in the description and Q&A tasks is difficult and challenging. To further verify the stability and bias of using commercial large language models as the evaluation criteria, we conducted multiple evaluation tests as well as tests using different commercial large language models for evaluation. The experimental results showed that the evaluation results of the three rounds were very similar, and the calculated variance was extremely low (all below 1%), which proved that the GPT-judge evaluations have good repeatability. In the description generation task, o4-mini, GPT-5, DeepSeek-reasoner, and DeepSeek-chat demonstrated similar prediction results, which proved the superiority of our model over Gemini-2.5-pro in this task. However, in the Q&A tasks, o4-mini and GPT-5 deemed our model as better, while DeepSeek-reasoner and DeepSeek-chat thought that Gemini-2.5-pro had a greater advantage. These new additional results have now been added in the supplemental information of our paper.
>
> 2. In description generation and Q&A tasks, the performance gap versus Gemini-2.5-Pro is small, which suggests that the overall advantage of L2M3OF remains modest compared to large-scale commercial models.
>
> We agree with the reviewer’s comment. As our response we have conducted further statistical testing to explore the statistical significance of our model’s superiority on some knowledge generation tasks. The statistical test revealed that L2M3OF consistently outperforms Gemini on the description generation task with a p-value of 0.00004 at a 95% confidence level with a Bonferroni correction (Bonferroni, 1936) due to the multiple judges, but not on the Q&A task with a p-value of 0.307, yet still outscoring Gemini, on average, according to o4-mini and GPT-5 ‘judges’. This further validates the reviewer’s comment and highlights the remarkable capabilities of rapidly evolving commercial LLMs on scientific tasks. Meanwhile we feel that this also underpins the great importance of competitively performing open-source models like our contributions.

---

> ### Author Response · Authors · 2025-12-03
> **Point-by-point replies to the comments2**
>
> 3. The paper does not report inference or fine-tuning cost. The L2M3OF’s multimodal architecture (PMTransformer + bridge + LLM) likely incurs higher computational overhead than text-only baselines (like L2M3OF).
>
> We thank the reviewer for pointing out this omission on our behalf. In fact, L2M3OF achieves a significant compression of the tokens in the crystal representation. The number of tokens required to represent a MOFs material through crystallographic information files is often between 10^3 and 10^4, while through the multimodal architecture's bridge, the number of tokens can be compressed to within 100. Therefore, the fine-tuning in GPU hours of L2M3OF (25.87) is significantly shorter than that of L2M2OF (551.29). We have not added this information in Section 4.1.
>
> 4. Have the authors experimented with training the model with LLM frozen? I suppose this would isolate the contribution of SFT on the LLM, and also tells us how much gain comes from multimodal alignment alone vs. from updating the language model. Especially, the work is compared with baselines of closed-source LLMs whose parameters can't be fine-tuned.
>
> We really appreciate the reviewer’s constructive comment as their proposed experiment helped us understand better the benefits of multimodal alignment alone. We have conducted the proposed test and in the new version of the paper, we have added the related findings and accommodated discussions (see Table 3 and Fig. 13). The experimental results show that although the model without SFT significantly underperforms compared to the L2M3OF baseline, yet it outperforms Gemini-2.5-pro in the tasks of property and description generation. This test further highlights the significant importance of multimodal alignment for large language models to understand material structures, especially spatial structure information.
>
> 5. There are multiple existing works that train crystal GNNs on MOF data. Therefore, is there any specific reason to justify the choice of PMTransformer as the structure encoder? Also, how large is the impact of the model architecture choice on the performance of L2M3OF?
>
> PMTransformer is a famous crystal encoder, which pre-trained on 1.9 million hypothetical porous materials, including MOFs, COFs and zeolites demonstrating universal performance across different tasks. PMtransformer (https://doi.org/10.1021/acsami.3c10323) has been shown to exhibit significant superiority over graph neural networks such as CGCNN and MEGNET which justifies our default choice for this transformer architecture.

---

### Official Review · Reviewer_nSw8 · 2025-10-31

**Soundness:** 3
**Presentation:** 3
**Contribution:** 3
**Rating:** 6
**Confidence:** 4

**Summary:**

This paper introduces MOF-SPK, a novel curated dataset comprising structural, property, and domain-knowledge information for over 100,000 Metal–Organic Framework (MOF) structures. Building on this dataset, the authors propose two models: L²M²OF (a text-only model) and L²M³OF (a multimodal text–structure model). The authors demonstrate that both models outperform generalist large language models (LLMs) on tasks defined within the MOF-SPK benchmark.

**Strengths:**

The paper makes a significant contribution to the field generation through the introduction of the MOF-SPK dataset, which unifies multiple material property prediction, description, and application tasks into a single, well-curated resource. Such an integrated dataset has strong potential to support the development and benchmarking of domain-specific foundation models for materials science.

Another strength lies in the dual-model design: the authors present both a text-only (L²M²OF) and a multimodal (L²M³OF) variant, allowing an insightful comparison of modalities. The comprehensive evaluation against several general-purpose LLMs further highlights the benefits of domain specialization.

The manuscript is well-organized, clearly written, and easy to follow, with a logical flow from dataset construction to model development and evaluation.

**Weaknesses:**

Despite the paper’s strong concept, the experimental design raises some concerns, particularly regarding baseline selection.

[Major] The comparison is limited to generalist language models (DeepSeek, GPT, Gemini), which may not provide a fair assessment of the proposed models’ performance. Numerous specialized neural architectures [a, b, c, d] for material property prediction and generation have been developed in recent years, and including at least a subset of these as specialist baselines would significantly strengthen the paper’s empirical claims.

[Minor] The description of the experimental and data processing pipelines could be more transparent. In particular, the prompts used for DeepSeek-R1 to extract material-relevant information from scientific publications are not provided, also it is unclear whether the generalist models were evaluated in zero-shot, one-shot, or few-shot settings. While the authors mention plans to release the code and processed data upon acceptance, a high-level overview of the data annotation pipeline and generalist model evaluation methodology should be included in the Supplementary Materials. This would improve reproducibility and accessibility, especially for readers who may not inspect the released code and data in detail.

a.Language models can generate molecules, materials, and protein binding sites directly in three dimensions as XYZ, CIF, and PDB files, Flam-Shepherd et al.

b.MOFDiff: Coarse-grained Diffusion for Metal-Organic Framework Design, Fu et al.

c.MOFGPT: Generative Design of Metal-Organic Frameworks using Language Models, Badrinarayanan et al.

d.Multi-modal conditional diffusion model using signed distance functions for metal-organic frameworks generation, Park et al.

**Questions:**

In addition to the points mentioned above, I have the following specific questions:

The paper states that PMTransformer is used as the structural encoder, but it remains unclear what textual backbone architecture is used for the L²M²OF and L²M³OF models. Please clarify the choice of language backbone and its adaptation to the multimodal setting.

The process of extracting SMILES representations via MOFid requires further clarification. In Figure 7 (Supplementary Materials), some examples show SMILES with disjoint fragments, which appears unintuitive. Could the authors provide additional statistics or qualitative analysis on the rate and correctness of recovered SMILES, particularly cases with multiple disjoint fragments?

---

> ### Author Response · Authors · 2025-12-03
> **Point-by-point replies to the comments**
>
> We thank the reviewer for recognizing our contributions in introducing the comprehensive MOF-SPK dataset and developing both text-only and multimodal LLMs that outperform general-purpose models on MOF-specific property and knowledge tasks. Point-by-point replies to the comments of reviewer are detailed as follows:
>
> 1. The comparison is limited to generalist language models (DeepSeek, GPT, Gemini), which may not provide a fair assessment of the proposed models’ performance. Numerous specialized neural architectures [a, b, c, d] for material property prediction and generation have been developed in recent years, and including at least a subset of these as specialist baselines would significantly strengthen the paper’s empirical claims.
>
> As explained in our response to another reviewer bjVx, the main goal of our contribution is to provide an open-source MOF LLM that materials practitioners can use across a wide range of prediction and language tasks and especially broad knowledge generation. While we explicitly compare against specific properties, as they directly link to the LLMs understanding of MOFs’ structures, our contribution is not a state-of-the-art property prediction model. Nevertheless, we understand that readers might want to see how our model compares against specialist models and we have now added additional comparisons against domain-specific MOF models MOFTransformer and CGCNN in the property prediction. Compared to MOFTransformer, our model can serve as a universal property prediction model, which means that various property prediction tasks can be integrated into a single model. However, although our current model supports the extraction of MOFs' structure, it does not support the inverse design of materials. Therefore, it cannot be compared with models a, b, and c. The new experimental findings have now been added in the supplemental information of our revised manuscript.
>
> 2. In addition to the points mentioned above, I have the following specific questions:The paper states that PMTransformer is used as the structural encoder, but it remains unclear what textual backbone architecture is used for the L²M²OF and L²M³OF models. Please clarify the choice of language backbone and its adaptation to the multimodal setting.
>
> Thanks for pointing out this missing details. We provided implementation details in the experiments section. We elaborated on the specific backbone model (Qwen2.5-7B) we selected and the LORA-based fine-tuning process, so that readers can better understand and replicate our approach.
>
> 3. The process of extracting SMILES representations via MOFid requires further clarification. In Figure 7 (Supplementary Materials), some examples show SMILES with disjoint fragments, which appears unintuitive. Could the authors provide additional statistics or qualitative analysis on the rate and correctness of recovered SMILES, particularly cases with multiple disjoint fragments?
>
> The reviewer is correct in that our description was not clear enough here. Metal-organic framework materials are generally composed of metal nodes and organic ligands. Therefore, their SMILES expression would be "disjoint fragments of metal and ligands". We have now added appropriate and detailed descriptions in the revised manuscript for the benefit of the reader. We apologize for the misunderstanding.

---

### Official Review · Reviewer_iJEg · 2025-11-02

**Soundness:** 2
**Presentation:** 3
**Contribution:** 2
**Rating:** 2
**Confidence:** 4

**Summary:**

This paper proposes L²M³OF, a multimodal LLM for MOF property prediction and analysis, along with L²M²OF, a text-only variant. The models are evaluated on geometric property prediction, structure extraction, description generation, and question answering using a newly curated MOF-SPK dataset.

**Strengths:**

- The approach of applying language models to MOF analysis is reasonable and timely (but has been done before)
- The paper develops both multimodal and text-only variants for comparison

**Weaknesses:**

**1. Poor absolute performance on basic properties:** The model's performance on fundamental geometric properties is concerning. For accessible surface area, the MAE exceeds 250 m²/g (L²M³OF) and approaches 500 m²/g (L²M²OF). These errors are substantial and raise serious questions about practical utility. The model appears good only because of the baselines chosen.

**2. Inappropriate baseline selection:** The critical flaw is comparing exclusively against general-purpose LLMs (GPT-5, Gemini, DeepSeek) in zero-shot settings, which are obviously unsuited for this specialized task. To demonstrate meaningful impact, the authors must compare against:
- State-of-the-art domain-specific models for MOF property prediction (MOFTransformer, graph neural networks, based on optimized features RACs/MOFDescribe)
- For the recommendation task, recent work from domain experts (e.g., Bernd Smit's group on MOF recommendations https://pubs.rsc.org/en/content/articlelanding/2024/dd/d4dd00116h)

**3. Limited property scope:** The evaluation focuses on relatively simple geometric properties (surface area, pore volume) rather than practically important properties like gas uptake capacity, selectivity, or other performance metrics relevant to the applications (carbon capture, hydrogen storage) claimed in the introduction.

**4. Minor nomenclature issue:** "L2M3” is already an established conference name in this field—suboptimal naming choice.

**5. CIF Performance in linker extraction** The advantage for CIF as input might be because CIFs are often sorted (metals first) which could simplify the task.

**Questions:**

1. **Why were no domain-specific MOF models included as baselines?** The paper cites MOFTransformer, DeepSorption, and other specialized models in the related work, yet only compares against general-purpose LLMs. Can you provide direct performance comparisons against these established MOF-specific methods?

2. **What is the practical significance of these error levels?** An MAE of 253.7 m²/g for surface area or 0.55 Å for PLD—are these acceptable for real MOF design workflows? What are typical ranges for these properties, and how do these errors compare to experimental measurement uncertainty?

3. **Can the model predict functionally important properties?** The evaluation focuses on geometric descriptors, but applications like carbon capture and hydrogen storage depend on gas uptake capacity, selectivity, and stability. Can you demonstrate performance on these practically relevant properties?

4. **How does performance compare to Bernd Smit's group's work on MOF recommendations?** You mention this as missing related work—can you include this comparison for the recommendation task?

---

> ### Author Response · Authors · 2025-12-03
> **Point-by-point replies to the comments1**
>
> We thank the reviewer for recognizing our work in developing both multimodal and text-only LLM variants for MOF property prediction and analysis, enabled by our newly curated MOF-SPK dataset. Point-by-point replies to the comments of reviewer are detailed as follows:
>
> 1. Poor absolute performance on basic properties and what is the practical significance of these error levels?
> Thanks for pointing out this missing detail in the interpretation of our findings. In the new version of paper, we have now provided the coefficient of determination (R2) for the property prediction tasks to help better evaluate the prediction results (see attached figure https://1drv.ms/i/c/0555194efdcdc29e/IQAgfUr6v7vVTZjGS2k-fhz_AWwYGmSceFv_8x45I6VCj5o?e=QDGEbZ). From the scatter plot of the predictions and the label values, it can be seen that R2 for L2M3OF in the accessible surface area and pore limiting diameter reaches 0.90 and 0.91 respectively, demonstrating the excellent performance of L2M3OF in property prediction.
>
> 2. Inappropriate baseline selection and why were no domain-specific MOF models included as baselines?
>
> As explained in our response to another reviewer bjVx, the main goal of our contribution is to provide an open-source MOF LLM that materials practitioners can use across a wide range of prediction and language tasks and especially broad knowledge generation. While we explicitly compare against specific properties, as they directly link to the LLMs understanding of MOFs’ structures, our contribution is not a state-of-the-art property prediction model. Nevertheless, we think it is very exciting that L2M3OF is yet very competitive on property prediction tasks against leading MOF predictive models, on top of its other general knowledge generation capabilities. However, to accommodate the reviewer’s comment here, we have now additionally compared our models against domain-specific models MOFTransformer and CGCNN in the property prediction. Among them, L2M3OF outperforms CGCNN in the prediction of pore limiting diameter, largest cavity diameter, accessible surface area and void fraction, and outperforms MOFTransformer in the prediction of accessible surface area. This demonstrates that our multimodal large language model L2M3OF has already approached, and in some cases even surpassed, specialized models specifically designed for property prediction in terms of property prediction tasks. The new findings have now been included in the Supplemental information of our revised manuscript.
>
> 3. Limited property scope and can the model predict functionally important properties?
>
> This comment overlaps with Comment 2 by Reviewer bjVx. In the new version of the paper, we have added up to 15 important high-level downstream tasks, such as band gap, gas adsorption and stability. Among the 15 downstream tasks, L2M3OF outperformed L2M3OF in 14 tasks, especially in the gas adsorption task which is more dependent on material spatial perception ability. However, L2M2OF only outperformed L2M3OF in the band gap prediction task that mainly relies on local element information. Our model can serve as a universal and general property prediction model, which means that various property prediction tasks can be integrated into a single model. The updated findings have been included in the experiments section of our revised manuscript.

---

> ### Author Response · Authors · 2025-12-03
> **Point-by-point replies to the comments2**
>
> 4. How does performance compare to Bernd Smit's group's work on MOF recommendations?
>
> Berend Smit’s work (https://pubs.rsc.org/en/content/articlelanding/2024/dd/d4dd00116h) is a MOF recommendation system via unsupervised Doc2Vec model trained on document-structured intrinsic MOF characteristics. However, there is a fundamental difference between the two works, i.e., their model recommends appropriate materials based on the application, while our model recommends an application for a specific material. Importantly, our model has a broader predictive capacity expanding to knowledge generation. Given that the work by Berend Smit's group only focuses on adsorption and quantum properties, which we have not trained our models on, a direct comparison is not possible. Nevertheless, we agree that this is indeed an importance refence linked to our work and we have now included it in our revised version.
>
> 5. Minor nomenclature issue: "L2M3” is already an established conference name in this field—suboptimal naming choice.
>
> We disagree with the reviewer. To the best of our knowledge there has only been organized one standalone workshop in the past with the L2M3 name, which is not even established. Our name choice is clearly distinct having superscripts and extra letters.
>
> 6. CIF Performance in linker extraction The advantage for CIF as input might be because CIFs are often sorted (metals first) which could simplify the task.
>
> We agree with the reviewer's opinion that in the structure extraction task, the elements represented by CIFs are more conducive for large language models to extract information about metals and linkers.

---

### Official Review · Reviewer_bjVx · 2025-11-04

**Soundness:** 1
**Presentation:** 3
**Contribution:** 1
**Rating:** 2
**Confidence:** 3

**Summary:**

This paper introduces a new model for the analysis of metal organic framework materials. The authors make several contributions in this work. First, they introduce a new dataset that they call MOF-SPK or "metal organic framework structure property knowledge". The dataset takes 133,000 materials from Cambridge Crystallographic Data Centre, and performs preprocessing such as deduplication. The materials are then labeled with computed properties such as density, pore limiting diameter and accessible surface area, as well as higher level properties such as the structure and information about applications characterization methods and stability extracted from a corpus of publications on the materials using DeepSeek-R1. The authors use this dataset to fine tune a large language model for predicting the above mentioned properties and attributes, with the input being a concatenation of a CIF-file representation of a given MOF with a prompt for a specific query. This basic purely-LLM based model is then extended by using a crystal encoder network (PMTransformer) to extract a representation of the crystal, another "bridge network" is used to transform this representation into tokens to use in place of the CIF file for the LLM input. The bridge network and LLM are fine-tuned together. In their evaluations, the authors compare their model to a range of state of the art LLMs for property-prediction, structure prediction and description tasks.

**Strengths:**

**Goals and novelty**
- The paper tackles an interesting and potentially important task of serving as an AI assistant for researchers investigating MOFs
- The approach of using a structure encoder to extract material representations combined with a natural language LLM for parsing questions is interesting and to my knowledge novel.

**Dataset**
- The dataset curated by the authors is potentially useful for other researchers in this area


**Writing**
- The writing of the paper is generally clear and the figures are generally clear and useful
- The related works are very up-to-date with recent developments in the field, helping put this work clearly into context.

**Weaknesses:**

**Novelty**
- The L2M2OF model is just a fine-tuned LLM, which is not particularly novel from a machine learning perspective
- The L2M3OF model underperforms L2M2OF across many tasks and its unclear if it's actually better on many of the higher level tasks.

**Evaluation**
- The only baseline are non-fine tuned LLMs
- The description and Q&A tasks are evaluated using another non-fine-tuned LLM as the judge, which is potentially biased and inaccurate as a metric.
- The ground truth labels for these tasks were also extracted with an LLM. That none of these steps are verified by domain experts is quite concerning
- The judgements compared to Gemini are quite close and it's not clear if they are statistically significant

**Justification**
- My understanding is that properties like LCD, PLD, density and SMILES representation can be directly and reasonably efficiently computed from the original CIF file representation (this is how the authors computed the ground truth labels). It unclear why it is useful to have an LLM produce potentially error prone predictions

**Missing details**
- Figure 1 suggests the model is using LORA-based fine-tuning, but this is not discussed anywhere in the text.
- No discussion of the computational costs, particularly in comparison to explicit property computation.

**Questions:**

Line 322: Why does the total number of tokens remain unchanged if more questions are being used in the input? Is it just that the crystal representation is shared across all of the questions?

---

> ### Author Response · Authors · 2025-12-03
> **Point-by-point replies to the comments1**
>
> We thank the reviewer for recognizing our contributions in constructing a large MOF structure–property–knowledge dataset and integrating crystal and language representations to enable effective multimodal MOF analysis. Point-by-point replies to the comments of reviewer are detailed as follows:
>
> 1. The L2M2OF model is just a fine-tuned LLM, which is not particularly novel from a machine learning perspective
>
> We disagree with the reviewer’s statement. While our clear contribution is the development of a multimodal LLM for a particular and very important scientific application, falling in the field of AI4Science, the architectural design features some important new designs that are crucial for the competitive performance of our model on such a challenging scientific problem. In particular, our architecture features a projection bridge, as well as multi-task training which might be common in multimodal systems, but their value in the modelling of crystalline materials had not been demonstrated. Beyond these we also facilitate a lightweight conversational data-augmentation scheme that concatenations multiple Q&A pairs in one batch to improve contextual diversity without increasing token counts. Our work provides the first clear evidence that these techniques are effective for feasibly capturing global-vs-local physical information of MOF structures. In addition, the group-training strategy offers a simple yet effective way to enrich contextual mixing, and it suggests that L2M3OF could fuse multiple crystal structures within a single conversation—highlighting a promising direction for in-context learning that we leave for future exploration. While all these architectural characteristics are not entirely novel, their strategic coupling is new and addresses critical challenges in crystalline materials machine-learned representation.
>
> 2. The L2M3OF model underperforms L2M2OF across many tasks and its unclear if it's actually better on many of the higher level tasks.
>
> As explained in Section 4.2 the original paper, the featurization of input materials can have different impact in the model performance when using CIF inputs versus transformer embeddings. The ordered structure of CIFs and sequential processing of LLMs brings up more local pattern emphasis in L2M2OF, and hence performs better across local substructure-induced properties, compared to the more ‘global’ inherent representation of PMTransformer embeddings that performs better on “higher level” tasks, yet being still quite competitive on local tasks. Nevertheless, to further assess the universal property performance of L2M3OF on more “higher level” tasks, including band gap, gas adsorption and stability. Among 15 different downstream tasks, L2M3OF outperformed L2M2OF in 14 tasks, especially in the gas adsorption task which is more dependent on material spatial perception ability. However, L2M2OF only outperformed L2M3OF in the band gap prediction task that mainly relies on local element information. Our model can serve as a universal and general property prediction model, which means that various property prediction tasks can be integrated into a single model. The following radar plot (https://1drv.ms/i/c/0555194efdcdc29e/IQBoN5O_awe3RYg1V6Ik0B37AfboObt8AdhwtuyfDhLHWZM?e=3g7xvG) illustrates the performance difference between the two proposed models. These additional results have now been added in the updated manuscript.
>
> 3. The only baseline are non-fine tuned LLMs
>
> Our main aim is to contribute a general-purpose, open, accessible chemistry tool for MOF prediction that goes beyond mere property prediction and generalizes to higher level concepts such as functionality and potential applicability. In that respect, the most accessible alternatives are commercially available LLMs which are often also fine-tuned on generic scientific tasks and this is why we compared only against them. Nevertheless, we understand that a natural query stemming from our comparisons could be on the head-to-head performance on other property specialized models, such as transformers and GNNs. The Table 4 in the new version of the paper demonstrates the performance comparison against two additional models trained on the same train-test split used for other LLMs, namely CGCNN and MOFTransformer. While the latter demonstrates the best overall performance in property prediction, our L2M3OF maintains second-best performance while also having the capacity to reason on higher level concepts around MOFs, offering not only a competitive predictor, but a general-purpose, accessible information tool that materials practitioners can easily interact with in the lab.

---

> ### Author Response · Authors · 2025-12-03
> **Point-by-point replies to the comments2**
>
> 4. The description and Q&A tasks are evaluated using another non-fine-tuned LLM as the judge, which is potentially biased and inaccurate as a metric.
>
> While this can be the case, we argue that evaluation by humans at such a large-scale is impossible and can be biased as well due to tiredness during evaluation and depending on the person’s chemical background. In response to the reviewer’s comment and to further verify the stability and bias of using commercial large language models as the evaluation criteria, we conducted multiple evaluation tests as well as tests using different commercial large language models for evaluation, namely GPT-5 and DeepSeek. In the description generation task, all three ‘judge’ models show agreement, while in the Q&A task the DeepSeek models deem the Gemini performance superior. We have now accommodated these new results in the main paper as an extra reassurance on the evaluation of our model.
>
> 5. The ground truth labels for these tasks were also extracted with an LLM. That none of these steps are verified by domain experts is quite concerning.
>
> Again, we argue that evaluation by humans at such a large-scale is impossible and can be biased. Nevertheless, we acknowledge the reviewer’s concern and to address that we have now used a stronger verified approach to generated ground truth labels for our training tasks. We specifically followed the approach described in https://pubs.acs.org/doi/10.1021/jacs.5c11789 (MOF-ChemUnity) which extracts verified MOF information from literature.
>
> 6. The judgements compared to Gemini are quite close and it's not clear if they are statistically significant
>
> While for some of the judgements, e.g., the description generation, the performance difference (in favor of L2M3OF) is statistically significant due to the large scale of description generation task the models were assessed, we agree that is not the case for all judgements such as the Q&A task, especially after adding new ‘judge’ LLMs to mitigate bias factors. Nevertheless, to properly respond to the reviewer’s comment with regards to statistical significance, we have performed two sign tests, one on the description generation task and one on the Q&A task, to assess statistical difference in the performance of our L2M3OF and Gemini. The statistical test revealed that L2M3OF consistently outperforms Gemini on the description generation task with a p-value of 0.00004 at a 95% confidence level with a Bonferroni correction (Bonferroni, 1936) due to the multiple judges, but not on the Q&A task with a p-value of 0.307, yet still outscoring Gemini, on average, according to o4-mini and GPT-5 ‘judges’. The latter demonstrates the remarkable capabilities of rapidly evolving commercial LLMs on scientific tasks – yet underpins the importance of competitively performing open-source models like ours.

---

> ### Author Response · Authors · 2025-12-03
> **Point-by-point replies to the comments3**
>
> 7. My understanding is that properties like LCD, PLD, density and SMILES representation can be directly and reasonably efficiently computed from the original CIF file representation (this is how the authors computed the ground truth labels). It unclear why it is useful to have an LLM produce potentially error prone predictions.
>
> A main goal of our contribution is to provide an open-source MOF LLM that materials practitioners can use across a wide range of prediction and language tasks, as an assistive tool in their day-to-day research activities. Many of the language tasks, e.g., Q&A, description and applicability generation are directly linked or strongly correlated to properties like LCR, PLD etc., and testing our models on these reveals its capacity to understand crucial aspects of the physical 3D structures of MOFs. Extracting SMILES representations of metals and ligands is a key task for testing whether the model can understand the local chemical information of MOFs materials. All these tasks together form an essential learning capacity that MOF LLMs need to have in order to produce sensible knowledge and answer questions on new MOF structures. Additionally, through these two benchmark tasks, we also explored the advantages and disadvantages of multimodal large language models compared to traditional large language models in understanding the structural space of MOFs materials. As we feel that this might not have been entirely clear in the previous version of our manuscript, we have now updated the Experimental setup section of our revised paper to convey this critical contribution motivation.
>
> 8. Figure 1 suggests the model is using LORA-based fine-tuning, but this is not discussed anywhere in the text.
>
> We thank the reviewer for pointing this crucial missing detail out. We have now added implementation details in the experiments section of the updated manuscript. We elaborated on the specific backbone model (Qwen2.5-7B) we selected and the LORA-based fine-tuning process, so that readers can better understand and replicate our approach.
>
> 9. Line 322: Why does the total number of tokens remain unchanged if more questions are being used in the input? Is it just that the crystal representation is shared across all of the questions?
>
> Thanks for pointing out the potential misunderstanding. The total number of tokens remain unchanged because we first sample a fixed batch of N instruction-answer pairs, where N is the declared batch size. After batching, we randomly group these instruction-answer pairs into groups of size 1,2,4,8,16 pairs and concatenate them to form multi-turn contexts. This grouping operation only changes how the questions are combined, not the total number of tokens in the batch—because the same N questions (and thus the same tokenized content) are still processed together. As a result, the effective batch size seen by the forward function varies depending on the grouping, but the total token count within the batch remains unchanged. To make this point clearer, we rephrased and added some explanation in Section 3.3.

---

### Official Review · Reviewer_1G3T · 2025-11-04

**Soundness:** 2
**Presentation:** 2
**Contribution:** 3
**Rating:** 4
**Confidence:** 4

**Summary:**

This paper introduces L2M3OF, a multimodal large language model for metal–organic frameworks (MOFs). It combines structural, textual, and domain-knowledge modalities to enable reasoning over 3D crystalline materials—an area where text-only models fail to capture reticular geometry and symmetry. The proposed model integrates a crystal encoder (PMTransformer) with a Qwen2.5-based LLM, linked by a projection and compression bridge that transforms geometric embeddings into token-level representations. Training is performed on MOF-SPK, a newly curated structure–property–knowledge dataset of 133,737 experimentally reported MOFs with associated literature.

**Strengths:**

The strengths of this work are as follows below:

Addresses a neglected modality gap, integrating structure and language for reticular materials.

Substantial new dataset (MOF-SPK) of >133k entries with curated properties and literature links.

Systematic evaluation across four tasks with both open and closed LLMs.

Methodologically clean hybrid (frozen encoder + lightweight bridge).

Joint-training ablation (Table 3) convincingly shows cross-task synergy.

**Weaknesses:**

The weaknesses of this work are as follows below:

There are reproducibility issues because dataset and code are not released; MOF-SPK curation process is only briefly described.

There is evaluation bias in comparing to closed models (GPT-5, Gemini) without uniform prompt design or temperature settings limits validity.

Because the projection-bridge multimodal alignment is standard, there is little architectural innovation beyond dataset scale.

The writing of the work can also be improved to more clearly and concisely deliver ideas.

Further, there are also critical references missing from this work. These include the below:

- Le, Khiem, Zhichun Guo, Kaiwen Dong, Xiaobao Huang, Bozhao Nan, Roshni Iyer, Xiangliang Zhang, Olaf Wiest, Wei Wang, Ting Hua, and Nitesh V. Chawla. MolX: Enhancing Large Language Models for Molecular Understanding with a Multi-Modal Extension. Proceedings of the ACM SIGKDD International Conference on Knowledge Discovery & Data Mining (MLoG-GenAI@KDD ’25), ACM, 2025.

- Guo, Zhichun, Kehan Guo, Bozhao Nan, Yijun Tian, Roshni G. Iyer, Yihong Ma, Olaf Wiest, Xiangliang Zhang, Wei Wang, Chuxu Zhang, and Nitesh V. Chawla. “Graph-based Molecular Representation Learning.” Proceedings of the International Joint Conference on Artificial Intelligence (IJCAI), 2023, pp. 6638-6646.

- Zeng, Zequn, et al. “MolXPT: Wrapping Molecules with Text for Generative Pre-Training.” ACL Short Papers, 2023.

- Soares, Eduardo A., et al. “An Open-Source Family of Large Encoder–Decoder Foundation Models for Chemistry.” Communications Chemistry, 8 (1), 2025.

- Kang, Yeonghun, et al. “A Multi-Modal Pre-Training Transformer for Universal Transfer Learning in Metal–Organic Frameworks.” Nature Machine Intelligence, 5 (3), 2023.

- Badrinarayanan, Srivathsan, et al. “MOFGPT: Generative Design of Metal–Organic Frameworks Using Language Models.” J. Chem. Inf. Model., 65 (17), 2025.

**Questions:**

How was dataset overlap between CCDC-derived structures and MOF-SPK evaluation sets prevented?

Can you quantify the data/computation cost of training vs GPT-5 baseline in FLOPs or GPU hours?

Did you test whether the frozen PMTransformer encoder limits adaptation to novel topologies?

How sensitive are results to projection size (M tokens)?

What safeguards exist to avoid hallucinations in material property predictions?

Can the model handle non-MOF crystalline systems (e.g., COFs or zeolites)?

How reproducible are your GPT-judge evaluations in description generation?

**Details Of Ethics Concerns:**

The authors need to more formally provide a response to the above ethics concerns in their paper.

---

> ### Author Response · Authors · 2025-12-03
> **Point-by-point replies to the comments1**
>
> We thank the reviewer for recognizing the contributions of our work in addressing the gap between material structural representations and natural language in large language models, as well as in advancing the construction of multimodal datasets for materials science. Point-by-point replies to the comments of reviewer are detailed as follows:
> 1. There are reproducibility issues because dataset and code are not released; MOF-SPK curation process is only briefly described.
>
> The code and SPK dataset were indeed not made publicly available. Nevertheless, we had originally included a disclaimer statement in the reproducibility section to inform the readers that these will be made publicly available upon acceptance.
>
> 2. There is evaluation bias in comparing to closed models (GPT-5, Gemini) without uniform prompt design or temperature settings limits validity.
>
> Thanks for pointing out the missing details. In the new version of paper, we have supplemented the evaluation prompts (Fig. 10) and temperature setting. Since the temperature of o4-mini cannot be set, the default value has to be used. Therefore, in the evaluation, the temperature of o4-mini is also set to the default value (temperature=1).
>
> 3. Because the projection-bridge multimodal alignment is standard, there is little architectural innovation beyond dataset scale.
>
> We disagree with the reviewer’s statement. While our clear contribution is the development of a multimodal LLM for a particular and very important scientific application, falling in the field of AI4Science, the architectural design features some important new designs that are crucial for the competitive performance of our model on such a challenging scientific problem. In particular, our architecture features a projection bridge, as well as multi-task training which might be common in multimodal systems, but their value in the modelling of crystalline materials had not been demonstrated. Beyond these we also facilitate a lightweight conversational data-augmentation scheme that concatenations multiple Q&A pairs in one batch to improve contextual diversity without increasing token counts. Our work provides the first clear evidence that these techniques are effective for feasibly capturing global-vs-local physical information of MOF structures. In addition, the group-training strategy offers a simple yet effective way to enrich contextual mixing, and it suggests that L2M3OF could fuse multiple crystal structures within a single conversation—highlighting a promising direction for in-context learning that we leave for future exploration. While all these architectural characteristics are not entirely novel, their strategic coupling is new and addresses critical challenges in crystalline materials machine-learned representation.
>
> 4. The writing of the work can also be improved to more clearly and concisely deliver ideas.
>
> We somewhat disagree with the reviewer. We have invested a substantial amount of time and effort putting thought in the organization of our paper, as well as its communication to a non-chemistry audience to ensure the content will be suitable and beneficial to the ICLR community. In this we have put a lot of effort in the writing quality of our work to ensure messages are clearly communicated. Given that the reviewer does not supply any specific examples where text in the manuscript is unclear or concise, and given the polar opposite comment by Reviewer H4S1 highly praising the writing in our manuscript, we have decided to take no action on this comment. Yet we did carefully check the flow and clarity of our writing once again to ensure messages are clearly communicated to ICLR audience.
>
> 5. Further, there are also critical references missing from this work.
>
> We thank the reviewer for their suggestion for additional references pertinent to our work. We have now accommodated suggested references in our revised manuscript.

---

> ### Author Response · Authors · 2025-12-03
> **Point-by-point replies to the comments2**
>
> 1. How was dataset overlap between CCDC-derived structures and MOF-SPK evaluation sets prevented?
>
> In our original manuscript we explicitly explained how dataset overlap is prevented in Section 3.1, paragraph X. To reiterate, we prevent overlap via 1) structural de-duplication using StructureMatcher (distance/tolerance thresholds and chemical matching rules), a tool that we already explain in Section 3.1 and 2) a strict chronological split by deposition year.
>
> 2. Can you quantify the data/computation cost of training vs GPT-5 baseline in FLOPs or GPU hours?
>
> We have now added text in Section 4.1 of our revised manuscript that discusses this. Fine-tuning L2M3OF is substantially less computationally intensive (25.87 GPU hours) as opposed to L2M2OF (551.29 GPU hours) owing to the large number of tokens as presented in Fig. 5B (usually 10^3-10^4) required to process crystallographic information files of MOFs.
>
> 3. Did you test whether the frozen PMTransformer encoder limits adaptation to novel topologies?
>
> To address the reviewer’s comment, we test L2M3OF’s inference performance on “out-of-distribution” structures such as Covalent Organic Frameworks (COFs) demonstrating competitive inference performance across multiple properties (see results herein: https://1drv.ms/i/c/0555194efdcdc29e/IQAgfUr6v7vVTZjGS2k-fhz_AWwYGmSceFv_8x45I6VCj5o?e=QDGEbZ). Given that this naturally follows from the universal predictive capabilities of PMTransformer we feel that it falls outside our work and we only include these performance comparisons here. However, it the reviewer thinks it might be beneficial for the readers, we are happy to include them in supplemental information.
>
> 4. How sensitive are results to projection size (M tokens)?
>
> We thank the reviewer for their suggestion, and we have now investigated the sensitivity to projection size. The following Table demonstrates little sensitivity of the L2M3OF model across different projection sizes to many different predictive tasks. We feel this ablation study is informative of the architectural design in our models and we have now added this in the supplemental information Table 5.
>
> 5. What safeguards exist to avoid hallucinations in material property predictions?
>
> To mitigate hallucinations we ensure that property prediction prompts provide enough supporting chemical information on each property at hand and how it correlates to the materials structures. Additionally, we provide a ‘backbone’ output template that ensures consistent outputs formats and minimization of hallucinations.
>
> 6. Can the model handle non-MOF crystalline systems (e.g., COFs or zeolites)?
>
> We have addressed this question earlier in our response to comment 3.
>
> 7. How reproducible are your GPT-judge evaluations in description generation?
>
> Thank you for raising this point. We performed three independent evaluation repeats using o4-mini for description generation (see results herein: https://1drv.ms/i/c/0555194efdcdc29e/IQDiN_Z1G405TrB7d6rZWurAAReXFK9BdGsnDrXgwGicUkM?e=6ioU5A). The results across all rounds were nearly identical, with standard deviation consistently below 1%. This demonstrates that the GPT-judge evaluation is highly stable and exhibits strong repeatability.

---

### Author Response · Authors · 2025-12-03
**Global response**

We thank the Area Chair and all reviewers for their constructive feedback and insightful comments. In particular, we appreciate the explicit recognition of the conceptual strength and potential impact of our dataset, as well as the practical utility of our proposed model framework, especially in a very challenging, yet filled with potential field of metal-organic frameworks.

As our response, we have very carefully taken into account all major and minor comments from all the reviewers and invested a lot of time and effort in a very extensive revision of the entire manuscript, including additional experiments and improving the presentation of our work to accommodate reviewers’ comments in great detail. In the following, we carefully respond to every comment and explain in which part of the revised manuscript we accommodate the changes.

We sincerely hope that our revision will satisfy the reviewers.

---

### Meta-Review · Area_Chair_iYio · 2026-01-04

**Summary:**

This paper introduces L^2M^3OF, the first multimodal large language model for metal-organic frameworks (MOFs), integrating crystal representation learning with language understanding to process structural, textual, and knowledge modalities jointly. The authors curated the MOF-SPK dataset containing over 133,000 experimentally reported MOFs with associated properties and literature. The reviewers acknowledge the ambition of creating a domain-specific multimodal model and the value of the curated MOF-SPK database. However, despite substantial revisions, major reviewer concerns still included: the lack of clear evidence that the multimodal architecture provides a fundamental leap in reasoning beyond specialized models, concerns regarding the reliance on LLM-based evaluation for complex scientific tasks, and the marginal gains observed in several downstream prediction tasks compared to simpler baselines.

**Reviewer Concerns:**

Addressed concerns:
 - Baseline selection (All reviewers): Authors added comparisons with domain-specific models (MOFTransformer, CGCNN).
 - Limited property scope (Reviewers iJEg and bjVx): Authors expanded evaluation to 15 downstream tasks including gas adsorption, band gap prediction, and stability.
 - Evaluation bias (Reviewers H4S1 and 1G3T): Authors conducted multiple validation rounds with different LLM judges and statistical significance testing.

Outstanding concerns:
 - Marginal Gains: Table 5 shows that changing the number of M-tokens has little effect on the prediction accuracy of properties (even in some tasks, adding parameters can lead to performance degradation).
 - Loss of chemical granularity in multimodal representations (Reviewr bjVx): The model struggles to extract explicit chemical identifiers (like SMILES strings) from structural embeddings as effectively as text-based models.
 - Limited evidence of added practical utility: Although additional tasks were introduced, the paper still does not clearly demonstrate scenarios in which the proposed model provides advantages over established methods commonly used in MOF research.

**Reviewer Scores:**

- Reviewer 1G3T (original score: 4): Would likely maintain at 4.
 - Reviewer bjVx (original score: 2): Would likely maintain at 2.
 - Reviewer iJEg (original score: 2): Would likely maintain at 2.
 - Reviewer nSw8 (original score: 6): Would likely maintain at 6.
 - Reviewer H4S1 (original score: 8): Would likely maintain at 8.

---

### Decision · Program_Chairs · 2026-01-26

Reject